# OpenReview forum: "The Adaptive Interrogator: Detecting Trojan LLMs in Multi-Agent Systems via Evolved Conversational Strategies"
_ICLR.cc/2026/Conference — ICLR 2026 Conference Withdrawn Submission_

### Official Review · Reviewer_BUcx · 2025-10-25

**Soundness:** 2
**Presentation:** 3
**Contribution:** 2
**Rating:** 2
**Confidence:** 3

**Summary:**

This paper presents a Trojan detection system against multiple agents systems. Intuitively, they use an initial seed set of Trojans and start mutating them using an Evolutionary Algorithm (EA), which rephrases, combines, and changes the Trojans to generate new ones. This process is mainly guided by a Large Language Model (LLM) Judge, which evaluates the outputs of all agents to the same questions and assigns them different scores. The experiments are primarily conducted on four different LLMs with 7-8 billion parameters. The paper shows promising results in most cases, across various numbers of Trojans and evaluation metrics.

**Strengths:**

* The paper addresses an important topic, as the Trojan/backdoor detection problem is highly significant.
* The threat model is very realistic, assuming only API access.

**Weaknesses:**

1. **Evaluation:**
   - There is a lack of clarity regarding the relationship between the used Trojans and the seed set that the EA works on. Even in section 4.6, the paper mentions 'different Trojan types' but does not explain them in detail. It would be beneficial to include a more description of how these Trojans are created and compare them to the ones in the seed set.
   - If these Trojans have similarities with the seed prompts, i would suggest to test the approach on different Trojans, such as using extremely rare words or random sets of tokens as triggers.

2. **Uniform Evaluation Sets:**
   - The sets compared are always from the same LLM. It would be beneficial to try different LLMs together since not all agents will be based on the same LLM. I believe this is important given the 25% false positive rates shown in Table 5.  It would also be helpful if a description of Trojans are used for Table 5 is presented as i could not find it.

3. **Evaluation Against Adaptive Defenses:**
   - The paper should consider evaluating against adaptive defenses. If the attacker knows the set of seed Trojans and can construct a judge, would the attacker be able to bypass this defense? For example. a naive example would be to include a prompt injection in the output to fool the judge. At least a discussion about this case is needed i believe.

4. **Baseline Comparison:**
   - Is using random questions as the baseline really the best approach? There might be more effective baselines based on state-of-the-art detection techniques.
   - The paper claims relevance to multi-agent systems, but the detection system appears to work independently on each agent (with the exception of the judge). Wouldn't this mean that any backdoor detection technique could also work in this setting? Hence, why can't related works be evaluated on each agent independently?

### Potential Typos  (not a weakness at all but couldnt find other place to put them)

1. **Line 141-142:** In the summary part of the introduction, 'potential Furthermore' seems to be missing a period.
2. **Line 214:** "EA (EA)" should likely be "Evolutionary Algorithm (EA)" on the first occurrence.

**Questions:**

Beside the questions above i have the following one:

- It is unclear why the performance worsens with more agents (section 4.4, Table 3). If the system works better with batches of 4 agents, why not run against 4 agents independently? This might only affect the judge, right?

---

### Official Review · Reviewer_o1qL · 2025-11-05

**Soundness:** 2
**Presentation:** 2
**Contribution:** 2
**Rating:** 4
**Confidence:** 4

**Summary:**

The paper proposes CTUS, a framework that uses an EA to evolve conversational probes and a judge LLM to score them for harmfulness, coherence, and repetition, to unmask trojan LLMs embedded in MAS. Experiments across Llama-2/3.1, Gemma, and Mistral agents report good Acc@1 (often 70–90%) and large gains over random prompting, with RLHF-induced backdoors being the hardest to detect.

**Strengths:**

- Novel Method
- Well-Structured Manuscript

**Weaknesses:**

- Insufficient Experimental Coverage
- Unclear/Missing Description

**Questions:**

- Some LaTeX commands are not successfully identified and correctly displayed in the Abstract on OpenReview.
- Please provide one or more concrete real-world scenarios or case studies illustrating how such Trojan behaviors could lead to severe harms (e.g., data leakage, misinformation propagation, or unsafe actions) in deployed multi-agent systems?
- The "keyword-based trojan trigger dataset" is used to initialize the EA. However, no information has been provided about how the dataset was obtained or what it contains.
- The manuscript does not specify the implementation details for constructing the different trojan types (e.g., fine-tuning/poisoning), nor does it clarify whether the "keyword-based trojan trigger dataset" is actually used as the triggers for those trojans.
- Why is the default maximum number of iterations set to 200? This seems like an excessively large number. In addition, the impact of different maximum numbers of iterations on CTUS performance is not shown.
- All experiments appear to use homogeneous MAS where all agents share the same base LLM and the same trojan type. Could you demonstrate how CTUS would generalize to mixed-model and/or mixed-attack scenarios?
- Minors:
  - L214, "CTUS employs an EA (EA) that"
  - The format of table captions is not uniform. For instance, the captions of Tables 1&2 are aligned to the left, and the captions of Tables 3&45 are aligned to the center.
  - L358, "consistency efficacy"
  - L366, "RLHGF"
  - L456, "porbes"
  - Eq. 1, ";V;"
  - The cited paper for GPT-3.5 is the "GPT-4 Technical Report."
  - The year for the paper "Weight Poisoning Attacks on Pre-trained Models" should be 2020 rather than 2004.

---

### Official Review · Reviewer_JhKJ · 2025-11-05

**Soundness:** 2
**Presentation:** 3
**Contribution:** 2
**Rating:** 4
**Confidence:** 3

**Summary:**

This paper introduces a new framework for detecting trojaned LLMs within MAS. Unlike prior work that focuses on single-agent settings, CTUS addresses the unique risks posed by malicious LLMs embedded in multi-agent environments, especially when models are sourced from public repositories or accessed as black-box APIs. The core innovation is an evolutionary algorithm that enables a judge agent to automatically evolve conversational strategies, probing other agents to reveal hidden trojan behaviors. CTUS operates in a black-box fashion, requiring only API access, and is evaluated across several LLMs and various trojan attack types. The results demonstrate that CTUS can effectively and efficiently uncover trojaned agents, even in complex, real-world MAS scenarios, and outperforms random prompting baselines. The paper also provides extensive ablation studies on system size, judge model, and attack type, highlighting the robustness and generalizability of the approach

**Strengths:**

1. Originality. To the best of my knowledge, the studied problem, i.e., address the detection of trojaned LLMs in multi-agent systems, is a novel setting compared to prior work of single-agent scenarios.

2. Black-Box method: The method is designed to work without access to model internals, making it highly relevant for real-world deployments where only API access is available.

3. The authors conduct thorough experiments across multiple LLM architectures, trojan attack types (word-level, syntax-level, semantic-level, RLHF-based), and varying system sizes and densities of trojan agents. The proposed method shows clear improvements compared with random prompting baselines.

**Weaknesses:**

1. Insufficient Baseline Comparisons: The authors only compare against a naive "Random Prompting" baseline, achieving 7.20% accuracy versus CTUS's 79.11%. This comparison is less meaningful  as it lacks established trojan detection baselines. I wonder if existing black-box detection techniques  such as Neural Cleanse and BEAT
can be applied to this application and if they are comparable to CTUS.

2. Questionable Claims About Multi-Agent Focus: The paper claims to be "the first to harness multi-agent conversational evolution for backdoor detection," but their "multi-agent" setup is actually quite limited - it's essentially one EA-controlled agent probing individual target agents separately. True multi-agent trojan detection would involve analyzing emergent behaviors from agent interactions, which this work doesn't address.

3. Missing Ablation Studies: While the paper includes some ablations on judge models and system size, it lacks crucial ablations on fitness function components, mutation strategy effectiveness, and EA hyperparameter sensitivity.

**Questions:**

1. If I understand it correctly, the success of CTUS relies on the fact that the trigger can be detected by mutation strategy. If the trigger is some random string, does the method still work?

2. What is the value Tmin in this paper?

3. Since the results are average over 3 runs, I have concerns about the statistical significance given the stochastic nature of evolutionary algorithms. Could you provide error bars?

4. How does the algorithm distinguish trojan LLMs and less safety aligned LLMs. As shown in Harmbench [1], many popular LLMs can achieve high ASR even with direct requests.  For example Zephyr has a 83% ASR with direct prompting the harmful requests.

5.
I understand this is an empirical method. I would be good if any analysis or insights of why evolutionary approaches should be effective for this problem can be provided. I would also like to see discussions about search space characteristics, convergence guarantees, or sample complexity (This is an optional question, and I understand that the author may find this question difficult to answer).


[1] HarmBench: A Standardized Evaluation Framework for Automated Red Teaming and Robust Refusal

---

### Official Review · Reviewer_dytb · 2025-11-05

**Soundness:** 3
**Presentation:** 3
**Contribution:** 3
**Rating:** 6
**Confidence:** 3

**Summary:**

This paper proposes a novel black-box detection framework for Trojan detection in Multi-Agent Systems, which is a crucial and emerging research direction. The black-box scenario represents a highly practical detection approach. Extensive experiments demonstrate the superiority of the proposed method.

**Strengths:**

1.  The problem of detecting Trojan attacks in LLM Agents is highly critical.
2.  The black-box detection approach is very appealing and has significant practical relevance.
3.  The experimental validation covers a wide range of scenarios.

**Weaknesses:**

1.  **Heavy Reliance on LLM as a Judge:** We acknowledge the ablation studies conducted by the authors. However, a remaining question is whether the evaluation method of the "LLM as a Judge" is consistent with human evaluators. While I note that some existing work has demonstrated this consistency, appropriate human review would help to further alleviate my concern.
2.  **Computational Cost:** To my knowledge, evolutionary algorithms involve multiple iterations. This raises the question of whether the detection time and economic costs are excessively high, which might affect my judgment on the practicality of the detection method. This is not a fatal flaw, but providing quantifiable comparisons of the overhead would make the work much more appealing.
3.  **Generalization and Initial Population:** The initial population is initialized from a keyword-based dataset. In reality, this reliance on known words might limit the assessment of the work's generalizability, and it is difficult for a detector to acquire such information in a real-world setting. I would like the authors to address this. Furthermore, a clear threat model is missing, which makes it difficult for readers of a security paper to immediately grasp the specific scenario, and is one of the reasons for my concern about the initial population.
4.  **Limitation on RLHF Attacks:** I appreciate the authors' proactive acknowledgment of the limitations regarding RLHF effectiveness. However, RLHF attacks represent one of the most advanced and prevalent attack methodologies. This constitutes a significant limitation, though the work itself remains valuable. If the authors could resolve this limitation, the significance of the work would be further enhanced.

**Questions:**

1.  Add **human evaluation** to validate the "LLM as a Judge" mechanism.
2.  Provide a specific **analysis of experimental costs** (e.g., number of API calls, time).
3.  Add a **threat model** and address the concerns regarding the initial population.

---

### Official Review · Reviewer_J1Gz · 2025-11-08

**Soundness:** 2
**Presentation:** 3
**Contribution:** 2
**Rating:** 4
**Confidence:** 3

**Summary:**

This paper introduces the Conversational Trojan Unmasking System (CTUS), an Evolutionary Algorithm (EA)-based framework designed to detect trojans in multi-agent systems.

**Strengths:**

The writing is clear and easy to follow, the presentation is well-organized, and the idea is engaging.

**Weaknesses:**

It is unclear why detection methods designed for a single-agent setting, as well as most existing safety techniques such as prompt filtering, fail to work in this scenario.

The threat model is difficult to understand. Since the EA is derived from a keyword-based trojan trigger dataset (i.e., initial candidates include phrases or questions related to known trojan domains or suspicious keywords), it is unclear how such a dataset could exist or be obtained, especially considering that CTUS is intended as a pre-deployment screening tool and users typically have only black-box access to the model.

There is a minor typo: “EA(EA)” in line 214.

**Questions:**

NA

---

### Official Review · Reviewer_gx4e · 2025-11-11

**Soundness:** 2
**Presentation:** 3
**Contribution:** 2
**Rating:** 2
**Confidence:** 3

**Summary:**

The paper proposes the Conversational Trojan Unmasking System (CTUS), an evolutionary algorithm-based framework designed to detect Trojan LLMs in multi-agent systems (MAS). It functions as a black-box, pre-deployment screening tool by using a judge agent to score agent responses (based on harmfulness, coherence, and repetition) and guide the evolutionary algorithm in evolving effective conversational probes. The paper evaluates CTUS on Llama-2, Llama-3, Gemma, and Mistral against four attack types (word, syntax, semantic, and RLHF). The results show that CTUS achieves high detection rates (often 70–80% Acc@1) and is particularly strong against semantic attacks, though RLHF attacks remain challenging.

**Strengths:**

1. The paper works on the critical and timely security challenge of detecting Trojan LLMs within MAS. The proposed solution, CTUS, is appropriately framed as a black-box, pre-deployment screening tool, which is a practical and necessary approach for real-world scenarios involving third-party models.

2. The paper's core methodology, using an evolutionary algorithm to automatically evolve conversational probes, is a novel approach to this detection problem. This framework is supported by a comprehensive evaluation across multiple modern LLMs (Llama-2, Llama-3, Gemma, and Mistral) and a diverse set of four Trojan attack types.

**Weaknesses:**

1. While the paper frames the problem in a MAS context, its methodology is essentially single-agent detection. The framework fails to utilize or discuss any unique MAS properties, such as risks emerging from inter-agent interactions. CTUS probes each agent individually, a setup that does not reflect MAS-specific threats. This formulation is a significant weakness, as there are many existing works on single-agent or LLM detection, and the paper fails to differentiate its contribution from these established methods.

2. The paper claims a "black-box MAS" setting but grants CTUS unrealistic, privileged access. The ability to freely query every individual agent and observe their direct, isolated outputs contradicts the black-box assumption and is inconsistent with the challenge of evaluating a complex, interactive system.

3. The evaluation relies on a single, weak baseline: "Random Prompting." While CTUS predictably outperforms this baseline, this only proves that a guided search is superior to an unguided one. The paper omits comparisons against other sophisticated black-box optimization or red-teaming techniques, making it impossible to conclude that the proposed evolutionary algorithm is an effective or efficient method for this task.

4. The framework’s detection capability is fundamentally limited to a narrow class of Trojan payloads. The entire evolutionary process is guided by a fitness function that relies on a harmfulness score. As defined in the judge’s prompt, this score only rewards the detection of overtly unsafe or policy-violating content, such as hate speech. Consequently, CTUS is blind to stealthy but malicious payloads, such as data exfiltration, subtle bias injection, or silent functional triggers.

Minor Weakness:

1. The entire CTUS framework hinges on the unevaluated assumption that the Judge Agent is a reliable and accurate oracle. The Judge is a single point of failure, as it provides the sole fitness signal for the evolutionary search. The paper fails to analyze the Judge's potential for error, stochasticity, or its inability to perceive the very "nuanced, indirect" threats that CTUS claims to detect.

**Questions:**

1. Differentiate from single-agent or LLM detection methods.

2. Add stronger, adaptive baselines.

3. Expand the fitness function beyond harmfulness to enable the detection of stealthy payloads, such as data exfiltration.

---

### Note · Authors · 2025-11-30

I have read and agree with the venue's withdrawal policy on behalf of myself and my co-authors.